# Detection of SARS-CoV-2 in Terrestrial Animals in Southern Nigeria: Potential Cases of Reverse Zoonosis

**DOI:** 10.3390/v15051187

**Published:** 2023-05-17

**Authors:** Anise N. Happi, Akeemat O. Ayinla, Olusola A. Ogunsanya, Ayotunde E. Sijuwola, Femi M. Saibu, Kazeem Akano, Uwem E. George, Adebayo E. Sopeju, Peter M. Rabinowitz, Kayode K. Ojo, Lynn K. Barrett, Wesley C. Van Voorhis, Christian T. Happi

**Affiliations:** 1African Centre of Excellence for Genomics of Infectious Diseases, Redeemer’s University, Ede 23210, Osun State, Nigeria; ayinlaa@run.edu.ng (A.O.A.); ogunsanyao@run.edu.ng (O.A.O.); sijuwolaa@run.edu.ng (A.E.S.); saibum@run.edu.ng (F.M.S.); akanok@run.edu.ng (K.A.); george27@run.edu.ng (U.E.G.); sopejua@run.edu.ng (A.E.S.); happic@run.edu.ng (C.T.H.); 2Department of Biological Sciences, Faculty of Natural Sciences, Redeemer’s University, Ede 23210, Osun State, Nigeria; 3Center for One Health Research, Department of Environmental and Occupational Health Sciences, University of Washington, Seattle, WA 98109, USA; peterr7@uw.edu; 4Department of Medicine, Division of Allergy and Infectious Diseases, Center for Emerging and Re-Emerging Infectious Diseases (CERID), University of Washington School of Medicine, Seattle, WA 98109, USA; ojo67kk@uw.edu (K.K.O.); lynnbob@uw.edu (L.K.B.); wesley@uw.edu (W.C.V.V.)

**Keywords:** SARS-CoV-2, domestic animals, non-domestic animals, reverse zoonosis, surveillance, Nigeria

## Abstract

Since SARS-CoV-2 caused the COVID-19 pandemic, records have suggested the occurrence of reverse zoonosis of pets and farm animals in contact with SARS-CoV-2-positive humans in the Occident. However, there is little information on the spread of the virus among animals in contact with humans in Africa. Therefore, this study aimed to investigate the occurrence of SARS-CoV-2 in various animals in Nigeria. Overall, 791 animals from Ebonyi, Ogun, Ondo, and Oyo States, Nigeria were screened for SARS-CoV-2 using RT-qPCR (*n* = 364) and IgG ELISA (*n* = 654). SARS-CoV-2 positivity rates were 45.9% (RT-qPCR) and 1.4% (ELISA). SARS-CoV-2 RNA was detected in almost all animal taxa and sampling locations except Oyo State. SARS-CoV-2 IgGs were detected only in goats from Ebonyi and pigs from Ogun States. Overall, SARS-CoV-2 infectivity rates were higher in 2021 than in 2022. Our study highlights the ability of the virus to infect various animals. It presents the first report of natural SARS-CoV-2 infection in poultry, pigs, domestic ruminants, and lizards. The close human–animal interactions in these settings suggest ongoing reverse zoonosis, highlighting the role of behavioral factors of transmission and the potential for SARS-CoV-2 to spread among animals. These underscore the importance of continuous monitoring to detect and intervene in any eventual upsurge.

## 1. Introduction

Coronavirus disease 2019 (COVID-19) is an ongoing pandemic that has posed an extraordinary threat to public health globally. It is caused by the severe acute respiratory syndrome coronavirus 2 (SARS-CoV-2) [1,2]. In December 2019, SARS-CoV-2 emerged in Wuhan, China, and caused an outbreak of pneumonia characterized by fever, cough, chest discomfort, dyspnea, and bilateral lung infiltration [3,4]. Human-to-human spread of SARS-CoV-2 occurs primarily through contact with an infected person when they cough or sneeze, or through droplets of saliva or discharge from the nose [5].

SARS-CoV-2 is a single, positive-strand RNA virus belonging to the family Coronaviridae and genus betacoronavirus (β-CoV) [6]. Like other coronaviruses, SARS-CoV-2 crosses the species barrier into humans [7,8]. Presently in humans, over 762 million confirmed cases and 6.8 million deaths have been reported globally [9], and 266,665 cases and 3155 deaths in Nigeria [10]. However, it has been postulated that many COVID-19 cases went unreported in Nigeria and other countries [11,12,13]. Moreover, several studies have shown that most people infected with coronaviruses are asymptomatic [14,15]. Therefore, these numbers are very likely an underestimation of the true picture of the COVID-19 burden globally, and in Nigeria. Notably, Southern Nigeria has been categorized as a high-risk zone for SARS-CoV-2 transmission [16].

A majority of the initial cases of COVID-19 requiring hospitalization were epidemiologically linked to the Huanan Seafood Wholesale Market, a wet market in Wuhan, which sells seafood and live animals, including poultry and wildlife [17,18]. Therefore, it is not surprising that the involvement of animals, especially wildlife, in the emergence of COVID-19 has been suggested. The existence of RaTG13, a betacoronavirus that shared a 96% whole-genome sequence identity with SARS-CoV-2, and other SARS-CoV-2-related viral genome sequences in bats [19,20] suggests that SARS-CoV-2 originated in bats and that intermediate hosts exist in the COVID-19 transmission pathway [21]. 

As COVID-19 spread in many regions of the world, a different transmission dynamic; human-to-animal transmission, has been postulated [22]. This is because SARS-CoV-2 has been detected in dogs and cats from households with confirmed human cases of COVID-19 in Argentina, Brazil, China, Spain, and the United States [23,24,25,26,27]. It has also been detected in minks in Denmark, the Netherlands, and the US [28,29,30,31], in white-tailed deer in the US [32], and in pumas, and lions in South Africa and the US [33]. In Sub-Saharan Africa, the cohabitation of humans with companion and farm animals is not an uncommon phenomenon. This situation provides ample opportunity for exposure of these domestic animals to SARS-CoV-2 by contact with infected humans, which can result in reverse zoonosis [25]. However, there are gaps in the knowledge on the extent of SARS-CoV-2 spread to other animals in Africa; this underscores the need for more information to guide the adoption of effective public health measures. This is essential for implementing sustainable and holistic measures for monitoring and controlling the spread of the virus. Therefore, we conducted a study to screen domestic animals for infection with SARS-CoV-2 to further elucidate the maintenance and spread of this virus in Africa and Nigeria in particular.

The main objective of this study was to investigate SARS-CoV-2 in both apparently healthy and clinically ill domestic animals in Southern Nigeria. 

## 2. Materials and Methods

### 2.1. Ethical Approval

Ethical approval was obtained from the National Veterinary Research Institute (NVRI), Jos (AEC/03/120/22).

### 2.2. Study Setting

The samples were collected from domestic and non-domestic animals in close contact with humans from four (4) states in Nigeria including Ebonyi, Ogun, Ondo, and Oyo during the COVID-19 pandemic. Ogun, Ondo, and Oyo States are all close to Lagos State which was the epicenter of COVID-19 in Nigeria (Figure 1). Samples were collected between May and August 2021 and January and October 2022. Apparently healthy animals were defined as animals that showed no clinical signs suggestive of any disease. On the other hand, clinically ill animals were defined as animals presenting various clinical signs such as fever, anorexia, diarrhea, respiratory distress, and skin lesions. 

### 2.3. Study Design

A cross-sectional study design was employed to carry out this study. Various domestic animals found living in close contact with humans such as in households and small backyard farms (cats, cattle, chickens, dogs, ducks, goats, lizards (*Agama agama*), pigs, pigeons, sheep, and turkeys) were sampled. Samples collected for the zoonotic surveillance of Lassa fever in Southern Nigeria within the 2021 and 2022 period were included in the study. Whole blood was collected from all animals into sterile EDTA tubes except for the lizards, while oral and rectal/cloacal swabs (for lizards and birds) were collected from all animals in these locations. The swabs were collected into sterile commercial tubes containing 1 mL of viral transport medium (VTM). Swab samples collected from animals from Ondo, Ebonyi, and Ogun States were preserved into 1.5 mls DNA/RNA Shield^TM^ (ZYMO RESEARCH) for the monthly transportation to ACEGID. All the samples were transported to the site laboratory in cold chain for storage. Whole blood samples were immediately processed to obtain plasma and temporarily stored at −20 °C for 2 weeks in the site laboratory. They were subsequently maintained in a cold chain during transportation to the African Centre of Excellence for Genomics of Infectious Diseases (ACEGID) laboratory at Redeemer’s University, Ede, Nigeria where they were kept at −20 °C until they were analyzed. The samples were collected by trained veterinarians who wore their disposable suits, face masks, and hand gloves. 

### 2.4. RNA Extraction and Real-Time RT-qPCR Detection for SARS-CoV-2

Total RNA was extracted in batches of 12 samples using a QIAamp Viral RNA extraction kit (Qiagen, Hilden, Germany) according to the manufacturer’s instructions. Two negative controls (RNAse-free water) were included during RNA extraction of each sample set. Extracted RNA was stored at −20 °C until SARS-CoV-2 RT-qPCR was completed. 

Initially, about 100 samples were screened for SARS-CoV-2 in duplicate using the GeneFinder™ COVID-19 Plus RealAmp kit (OSANG Healthcare, Anyang-si, Republic of Korea), the AviMol Dri Kit (AVICENNA, Dubai, United Arab Emirates) and the Allplex™ SARS-CoV-2 plus Variants Assay kit (Seegene, Seoul, Republic of Korea) to validate the consistency of our results, each time starting the entire process from RNA extraction. After the validation process, the AviMol Dri kit showed consistent results and was found to be the most appropriate kit for the detection of SARS-CoV-2 in animals. Subsequently, it was used to analyze the rest of the samples.

Twenty-three (23 µL) microliters of thawed RNA, including the negative extracts and new set of two negative controls were vortexed, quickly spun, and added to tube strips containing the freeze-dried RT-qPCR Mix. The resulting mix was then transferred into a 96-well plate and run on a Roche LightCycler^®^ 96 machine. The cycling conditions include reverse transcription at 50 °C for 10 min, pre-denaturation at 95 °C for 3 min, 45 cycles each of denaturation at 95 °C for 10 s, and annealing and extension at 60 °C for 60 s. All steps were carried out in strict adherence to the AviMol Dri kit protocol produced by Avicenna (Dubai, United Arab Emirates). The kit is designed to amplify 3 target genes: ORF1ab gene, N-gene, and the RNase gene. Samples were considered positive based on two criteria: the cycle threshold (Ct) values being less than or equal to 38 for the N-gene and ORF1ab gene; the amplification curve being S-shaped for the N-gene and ORF1ab gene. Samples with Ct values between 38 and 40 were rerun and considered positive if the Ct values were less than 40 on the second test. An animal was considered positive if it was either positive using oral or rectal swabs or both. The RT-qPCR results were only true if the negative extracts and negative controls showed no amplification.

### 2.5. ELISA for Detection of Antibodies to SARS-CoV-2

ELISA was carried out on plasma samples from five domestic animal species (cattle, dog, goat, pig, sheep) for which anti-IgG-HRP was available as at the time this study was carried out. An indirect ELISA was used to detect antibodies (IgG) against SARS-CoV-2 in the domestic animal species using a previously described protocol with a few modifications [34]. Briefly, Immulon 4HBX Flat Bottom 96-well Microtiter Plates were coated with 100 µL of recombinant SARS-CoV-2 Nucleoprotein N-terminus amino acid 47-173 truncation (N-Nterm) (1:2400 diluted in sterile 1X PBS, final concentration 2 µg/mL) and Spike-Receptor Binding Domain (S-RBD) (1:250 dilution in sterile 1X PBS, final concentration 2 µg/mL) antigens. A sample dilution of 1:100 and species-specific anti-IgGs HRP dilution of 1:15,000 were used. Diluted samples were dispensed into wells in duplicates. The optical density (OD) was read at 490 nm using the Biotek 800 TS ELISA plate reader. Animals for which pre-COVID-19 plasma samples were available, the cutoff for positive versus negative was determined to be mean OD values of COVID-naive samples (pre-COVID-19 stored plasma) plus 3 times the Standard Deviation of the mean. Where pre-COVID-19 animal samples were not available (cattle, goat, pig, and sheep), we defined the cutoff as the mean OD of negative controls (pathogen-free sera from IDVet for respective animal species) plus four times the standard deviation. Samples with OD higher than the cut-off were recorded as positive for both N-Nterm and S-RBD-coated plates. 

### 2.6. Statistical Analysis

Data were documented serially using the unique animal identification number and were analyzed using version 8 Epi Info software (Centers for Disease Control and Prevention, Atlanta, GA, USA) and statistical program SPSS for Windows (version 25.0, SPSS Inc., Chicago, IL, USA). Simple graphs for results illustration were prepared using Graph Pad Prism version 8.0 and Microsoft Excel. Discrete variables were compared using test of proportion by calculating chi-square. Normally distributed, continuous variables were compared by Student’s *t*-tests and/or analysis of variance (ANOVA) as applicable. Data not conforming to a normal distribution were compared by Mann–Whitney U test or Kruskal–Wallis as applicable as well. All tests of significance were two-tailed and values of *p* < 0.05 were taken to indicate significant differences. 

## 3. Results

A total of 1279 samples from 791 animals were analyzed. The animals comprised 11 different species of apparently healthy (*n* = 751) and clinically ill (*n* = 40) animals. Goats were the most sampled animal (*n* = 234; 29.6%) followed by dogs (21.7%) (Table 1). With respect to location, most of the sampled animals were from Ondo (*n* = 381; 48.2%) (Table 1). Overall, 654 animals were screened using ELISA, 364 (623 samples) were screened using RT-qPCR and, out of these two groups, 227 were screened using both RT-qPCR and ELISA. 

In Ebonyi State, 243 animals were sampled: 184 were screened using ELISA and 106 using RT-qPCR. Goats were the most sampled and analyzed (38.7%) (Table 1). In Ogun State, 144 animals were sampled: 138 were screened using ELISA and 82 using RT-qPCR. Pigs were the most sampled and analyzed (80.6%) (Table 1). In Ondo State, 381 animals were sampled: 309 were screened using ELISA and 153 using RT-qPCR. Goats were the most sampled and analyzed (31.5%) (Table 1). Finally, in Oyo State, 23 animals were sampled and were all screened using ELISA and RT-qPCR. Pigs were the most sampled and analyzed (65.2%) (Table 1).

### 3.1. RT-qPCR Results for SARS-CoV-2 Detection in Animals

SARS-CoV-2 was detected in all animal species analyzed by RT-qPCR except pigeons. Of the 364 animals screened by RT-qPCR, 167 (45.9%) were confirmed positive for SARS-CoV-2. Positivity to SARS-CoV-2 considering both oral and rectal swabs was 16.5% (60/364). With oral swabs only, 39.9% (137/343) were confirmed positive while 32.5% (91/280) were positive with rectal swabs only. There was no significant difference in SARS-CoV-2 positivity between oral and rectal swab samples (*p* = 0.07). The Ct values of SARS-CoV-2 positive animals ranged from 28.52 to 39.37 using oral swabs and 24.99 to 39.31 using rectal swabs. Overall, there was no significant difference in the mean Ct values of oral and rectal swabs (*p* = 0.73). However, in cattle, the mean Ct value for rectal swabs was significantly higher than that of oral swabs (*p* = 0.048) (Figure 2).

#### 3.1.1. Comparison of SARS-CoV-2 RT-qPCR Positivity Rates by Location

Using RT-qPCR, SARS-CoV-2 was detected in animals in all sampled locations except Oyo State. Among the four states, Oyo State recorded a 0% positivity rate while Ogun State recorded a significantly lower positivity rate (4.9%) compared with Ebonyi and Ondo State (*p* < 0.0001) (Figure 3).

#### 3.1.2. Comparison of SARS-CoV-2 Positivity Rates by Animal Species per Location Using RT-qPCR

Overall, SARS-CoV-2 positivity rates were significantly higher in sheep (*p* < 0.0001), cattle (*p* = 0.01), and goats (*p* = 0.01) but were significantly lower in pigs (*p* < 0.0001) and chickens (*p* < 0.01) compared to other species sampled (Table 2).

In Ebonyi State, SARS-CoV-2 positivity rates were significantly higher in sheep (*p* = 0.001) and cattle (*p* = 0.01) but were significantly lower in lizards (*p* = 0.001) compared to other species (Table 2). In Ogun State, all animals analyzed had SARS-CoV-2 positivity rates of ≤20% (Table 2). However, there was no significant difference in positivity rates among the different species analyzed. In Ondo State, SARS-CoV-2 positivity rates were significantly higher in sheep (*p* < 0.0001) and lizard (*p* = 0.01) but were significantly lower in chicken (*p* < 0.0001) and turkey (*p* = 0.01) compared to other species (Table 2).

#### 3.1.3. Comparison of SARS-CoV-2 RT-qPCR Positivity Rates by Health Status

Of the 364 samples screened through RT-qPCR, 325 were obtained from apparently healthy animals and 39 from clinically ill animals. SARS-CoV-2 positivity was not significantly higher in clinically ill animals (59.0%) compared to the apparently healthy ones (44.3%) (*p* = 0.09). 

#### 3.1.4. Comparison of SARS-CoV-2 RT-qPCR Positivity Rates by Sampling Period and Location

Out of the samples screened using RT-qPCR, 183 were obtained in 2021 and the remaining 181 were obtained in 2022. SARS-CoV-2 positivity was significantly higher in animals sampled in 2021 (61.2%) compared to those sampled in 2022 (30.4%) (*p* < 0.0001). Samples from Oyo State were all obtained in 2022. The highest positivity rates (100%) were observed in May and June 2021 for both Ebonyi and Ondo States. This was followed by a decrease in positivity in July 2021 to 88.2% and 90.4% in Ebonyi and Ondo States, respectively. Despite the fluctuation in positivity rates in both states, SARS-CoV-2 positivity was higher in 2021 than in 2022 (Figure 4). In Ebonyi State, SARS-CoV-2 positivity dropped to levels lower than 20% from February while in Ondo State, it remained above 40% till the end of sampling in June 2022 (Figure 4).

### 3.2. ELISA Results of Domestic Animals Naturally Infected with SARS-CoV-2

Of the 654 animals screened using ELISA, 13 animals (2%) expressed IgG antibodies to the S-RBD antigen only while 14 animals (2.1%) expressed antibodies to the N-Nterm antigen only. Interestingly, nine animals (1.4%) comprising four goats and five pigs expressed IgG antibodies to both the S-RBD and N-Nterm SARS-CoV-2 antigens. Hence, these were confirmed positive SARS-CoV-2 using ELISA (Figure 5a). The cut-off OD values for the S-RBD plates were 0.154, 0.174, 0.16, and 0.156 for cattle, goats, sheep, and pigs, respectively. For the N-NTerm plates, the cut-off OD values were 0.152, 0.162, 0.162, and 0.164 for cattle, goats, sheep, and pigs, respectively. The OD values for SARS-CoV-2 positive animals ranged from 0.163 to 0.283 for S-RBD and from 0.166 to 0.292 for N-NTerm (Figure 5b).

#### 3.2.1. Comparison of SARS-CoV-2 ELISA Positivity Rates by Location

SARS-CoV-2 was detected in Ebonyi and Ogun States. The SARS-CoV-2 infection rate was higher in Ogun State (3.6%) compared to Ebonyi State (2.2%), but this was not statistically significant (*p* = 0.51).

#### 3.2.2. Comparison of SARS-CoV-2 ELISA Positivity Rates by Animal Species per State

Overall, the SARS-CoV-2 positivity rate was greater in pigs (3.6%) than in goats (1.8%), but this was not statistically significant (*p* = 0.31) (Figure 6a).

In general, SARS-CoV-2 positivity using ELISA was observed in one species per state. In Ogun State, pigs were the only animal species in which antibodies to SARS-CoV-2 were detected and in Ebonyi state, antibodies to SARS-CoV-2 were only detected in goats (Figure 6b).

#### 3.2.3. Comparison of SARS-CoV-2 ELISA Positivity Rates by Health Status

Of the 654 samples screened through ELISA, 618 were obtained from apparently healthy animals and 36 from clinically ill animals. Only the apparently healthy animals were SARS-CoV-2 positive by ELISA (1.5%) while none (0%) of the clinically ill animals were positive for SARS-CoV-2. 

#### 3.2.4. Comparison of SARS-CoV-2 ELISA Positivity Rates by Sampling Period and Location

Out of the samples screened using ELISA, 487 were obtained in 2021 and the remaining 167 were obtained in 2022. Antibodies to SARS-CoV-2 were detected only in animals sampled in 2021 (1.8%). Like in the case of PCR, the highest positivity rate (10.5%) to SARS-CoV-2 was observed in May 2021 (Figure 7).

### 3.3. Juxtaposition of RT-qPCR and ELISA Results

One pig sample from Ogun State was positive for both SARS-CoV-2 by RT-qPCR screening and antibodies against the S-RBD and N-NTerm antigens using ELISA.

## 4. Discussion

The origin of SARS-CoV-2 from animals and the zoonotic spillover of the virus have been documented [21,35]. There have also been reports of SARS-CoV-2 transmission from humans to pets such as dogs and cats in households with confirmed cases of SARS-CoV-2 [23,24]. This study presents a broader view as it extends to other animal species that are in proximity to humans based on community lifestyle (human–animal cohabitation) commonly practiced in many regions in Nigeria. In this study, we describe cases of reverse zoonoses of SARS-CoV-2 in domestic animals. We also present the highest infectivity rate of SARS-CoV-2 reported in domestic animals so far in Africa and the largest range of domestic animals ever surveyed for SARS-CoV-2. Our findings show a 45.9% SARS-CoV-2 positivity rate using RT-qPCR and a 1.4% rate using ELISA in the animals surveyed. It is anticipated that the high positivity rate recorded in domestic animals during this study raises the question of possible contamination during sampling or sample processing. During sampling, the samples were collected by the veterinary clinicians wearing personal protective gear (face mask, hand gloves, and disposable lab coat). The samples were collected directly into their respective sterile tubes. In the laboratories, both on the site and at ACEGID, samples were processed under the laminar flow hood periodically disinfected. However, PCR was only completed on swab samples which were collected into pre-labeled sterile tubes containing sterile VTM and directly stored without processing. Negative controls were included in the processes of RNA extraction and RT-qPCR preparation for analysis for a quality check to ensure that there was no sample contamination. 

Overall, there was no significant difference in SARS-CoV-2 positivity rates and cycle threshold values between oral and rectal swab samples among animals. This suggests that either of the samples can be used for an appropriate diagnosis of SARS-CoV-2 infection. However, using both samples increases the likelihood of detecting SARS-CoV-2 in animals.

SARS-CoV-2 was detected in all states sampled except Oyo State. This is probably because of the relatively smaller sample size (*n* = 23), and more importantly, all the samples from Oyo State were obtained in early October 2022, a period where the prevalence of COVID-19 had declined in Oyo State and Nigeria as a whole. According to the Nigeria Centre for Disease Control (NCDC), no COVID-19 cases were reported in humans in Oyo in early October 2022. Coincidentally using ELISA and RT-qPCR, SARS-CoV-2 was detected only and mostly in animals sampled in 2021, suggesting the transient period of protection against SARS-CoV-2 in animals. Additionally, there was an 81.7% reduction in the number of human COVID-19 cases from early October 2021 (1842 cases) to early October 2022 (337 cases) in Nigeria [10]. This suggests a link between human–animal SARS-CoV-2 transmission and reverse zoonosis cases. 

Our study demonstrates the ability of SARS-CoV-2 to infect a range of domestic animals. This is expected because coronaviruses can adapt to a wide range of hosts due to their ability to mutate very quickly [36]. All animal species included in this study except pigeons tested positive for SARS-CoV-2 using RT-qPCR. However, only one pigeon was sampled, and this sample size does not allow justifiable conclusions to be made about the SARS-CoV-2 status in pigeons in southern Nigeria. SARS-CoV-2 positivity rates were significantly higher in ruminants using PCR with the highest observed in sheep. This is in tandem with the findings of some authors who reported SARS-CoV-2 detection in ruminants [37,38,39,40]. Ulrich et al. [37], Bosco-Lauth et al. [38], and Fernández-Bastit et al. [39] reported RT-PCR-positive nasal samples in cattle and goats. In addition, Gaudreault et al. [40] demonstrated that sheep-derived kidney cells support SARS-CoV-2 replication and that viral RNA was also detected in sheep respiratory tract and lymphoid tissues. Given that in Nigeria and in Africa as whole, small ruminants, particularly goats, are kept in human dwellings it was not surprising to observe the significantly higher SARS-CoV-2 positivity rate in them compared to other animals tested. 

Although SARS-CoV-2 positivity rates were significantly lower in pigs and chickens in this study, it was detected by RT-qPCR in pigs and confirmed that they seroconvert using IgG ELISA. The detection of SARS-CoV-2 RNA in pigs and chickens is still in contrast with the findings of Schlottau et al. [41] and Suarez et al. [42] who reported no detection of viral RNA or seroconversion in experimentally infected pigs and chickens. Contrarily, other studies carried out later reported PCR-positive nasal/oral or rectal swabs and observed antibody response to SARS-CoV-2 in experimentally infected pigs [43,44]. Notably, the prevalence of SARS-CoV-2 in ruminants, pigs, and poultry was higher in our studies than in previous studies. However, these studies were carried out on experimentally infected animals whereas, in our study, the animals were naturally infected with SARS-CoV-2. 

There was no significant difference in SARS-CoV-2 positivity between apparently healthy and clinically ill animals. However, the clinically ill animals were sampled from Ondo State only while the apparently healthy animals were from Ebonyi, Ogun, Ondo, and Oyo States. Therefore, this does not allow for an appropriate unbiased comparison between the two animal groups. Nevertheless, it appears that health status does not influence susceptibility to SARS-CoV-2 in animals.

In Nigeria, and many other countries in Africa, clinical cases of COVID-19 in humans were relatively low in comparison to other continents [45]. However, our analysis shows that there was a high infection rate of SARS-CoV-2 in animals (61.2%) during the peak of the pandemic in 2021. This is likely due to their proximity to humans which allowed for an intense exposure of these animals to SARS-CoV-2-positive humans [25], subsequently leading to reverse zoonosis. For instance, in many households where samples were obtained for this study, animals were either tethered very close to the house or were from backyard farms close to the human dwellings. Again, small pig farms with regular contact between humans and pigs in Ogun State during feeding and cleaning suggest increased exposure of those animals to SARS-CoV-2-infected humans. The potential existence of many asymptomatic cases in humans [46] may have also contributed to the transmission of SARS-CoV-2 to domestic animals as the humans were not diagnosed with COVID-19. For all the animals in this study, there was a considerably lower SARS-CoV-2 infectivity rate in 2022 compared to 2021. We postulate that this is due to the reduction in the prevalence of human cases of COVID-19 in Nigeria in 2022. This finding also strengthens our argument that the SARS-CoV-2 infection detected in animals was likely transmitted from humans. 

Contrary to what was observed by RT-qPCR, a low percentage of the animals (1.4%) screened using ELISA were seropositive for SARS-CoV-2. This overall lower seropositivity to infection rate could signify that many animals become naturally exposed and shed SARS-CoV-2 in nasal and rectal secretions, yet most of these animals do not become sufficiently infected to stimulate IgG immunity or it could be that the animals are in their acute stage of infection. Pigs and goats expressed IgG against both S-RBD and N-Term SARS-CoV-2 antigens. Previously, Vergara-Alert et al. [44] detected low levels of antibodies directed against the SARS-CoV-2 Spike and N proteins. Although these animals were experimentally infected, it provides evidence of seroconversion in pigs. Importantly, 14 dog and goat samples expressed no antibodies against the S protein, but had antibodies against the N protein, suggesting a possible cross-reaction with other coronaviruses which may have infected these animals prior to the time of sampling. This may be because the N protein is relatively conserved among coronaviruses that infect animals and humans [47].

Although sheep had the highest positivity rate using RT-qPCR, there was no evidence of seroconversion to SARS-CoV-2 against the S-RBD and N-Term antigens in sheep. This suggests that the sheep sampled in this study were probably in the active phase of infection during the sampling period. The existence of an RT-PCR-positive pig who also developed IgG to SARS-CoV-2 S-RBD and N-Nterm antigens may suggest that the pig had already seroconverted but had not cleared the antigen at the time of sampling, or it might have been infected with a different SARS-CoV-2 variant after prior seroconversion to another virus variant. Historically, pigs have been implicated in the transmission of viruses to humans [48]. Moreover, they serve as natural hosts for other coronaviruses such as porcine delta coronavirus, porcine epidemic diarrhea virus, and porcine hemagglutinating encephalomyelitis virus [49]. Therefore, it is important to further investigate the cross-reactivity of SARS-CoV-2 antibodies to other coronaviruses in pigs.

Cases of natural SARS-CoV-2 infection have been reported in dogs and cats, but to the best of our knowledge, this is the first report of SARS-CoV-2 natural infection in poultry, pigs, and lizards. The relatively high PCR positivity rate in some animal species in our study may have been driven by horizontal transmission between these animals, especially in those intensively reared. This is not surprising as once a pathogen crosses from an animal to a human, it is likely going to cross from one animal type to another animal taxa. Importantly, different studies have identified direct contact as the most efficient route for animal-animal SARS-CoV-2 transmission [41,50].

We acknowledge that our study has some limitations. The protocol and kits used to carry out RT-qPCR were designed for human subjects. However, the results were interpreted using the Ct values which correspond to the N-gene which are specific for SARS-CoV-2. Additionally, the cross-sectional design of our study limits our investigation of the cause–effect relationships between the animal cases and human cases. However, all animals sampled were found living close to humans. Finally, at the time this study was carried out, we had access to secondary antibodies (IgG) for only five animal species. This hindered our ability to screen for antibodies to SARS-CoV-2 in other species. 

## 5. Conclusions

This study demonstrated evidence of natural infection with SARS-CoV-2 in a wide range of animal species existing in close contact with humans with the use of molecular and serological analyses. The degree of SARS-CoV-2 detection in animals was greater in 2021 than in 2022, correlating with a decrease in human infection rates over the same period. The proximity between humans and animals in this setting as well as the temporal correlation of positivity rates strongly suggest a human–animal transmission. However, there is a need for further research into the role of SARS-CoV-2-positive animals in the maintenance of the virus and transmission to humans and among animals. Continuous monitoring of the virus in animals and humans is recommended for preparedness in case of mutations that may increase the virulence and transmissibility of the virus as its moves from species to species.

## Figures and Tables

**Figure 1 viruses-15-01187-f001:**
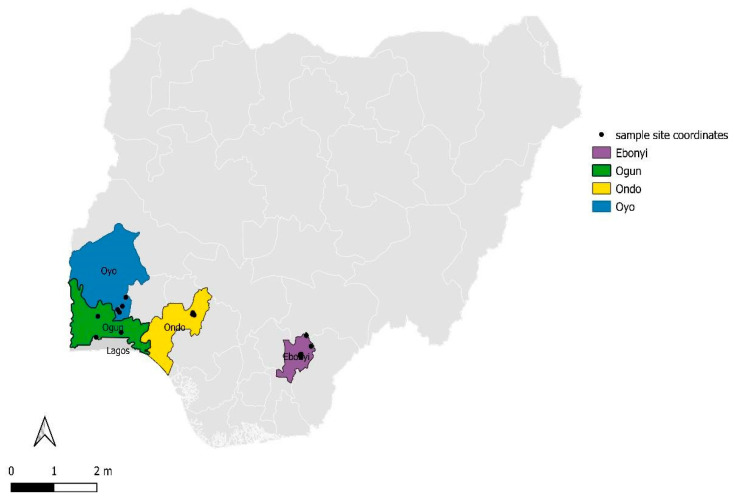
Map of Nigeria showing the sampling sites.

**Figure 2 viruses-15-01187-f002:**
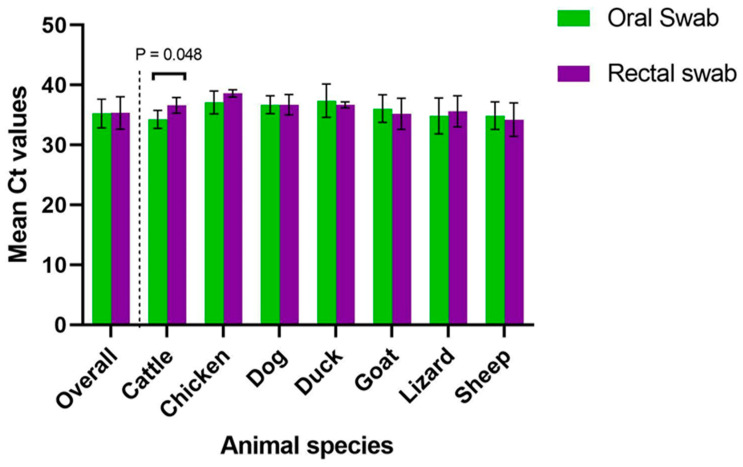
Comparison of PCR Ct Values in different animal species.

**Figure 3 viruses-15-01187-f003:**
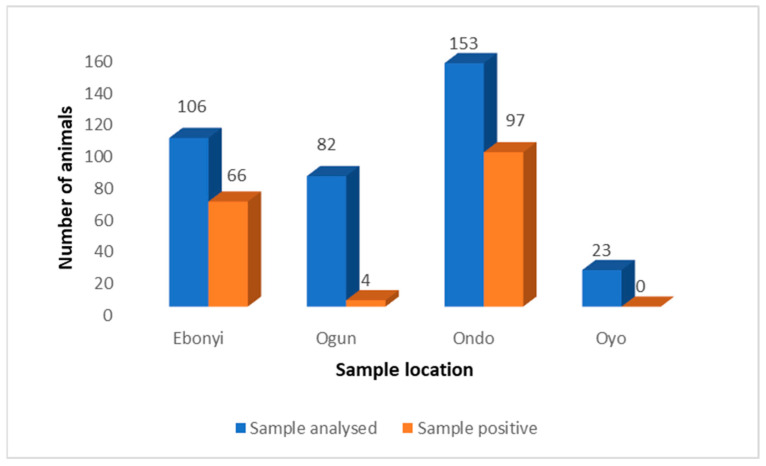
Comparison of SARS-CoV-2 PCR positivity rates in animals by location in Nigeria.

**Figure 4 viruses-15-01187-f004:**
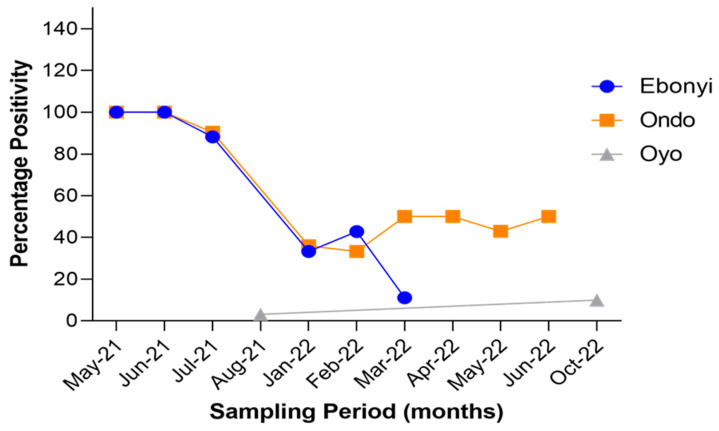
Temporal distribution of SARS-CoV-2 positivity using PCR in animal species in the sampling locations over the duration of study.

**Figure 5 viruses-15-01187-f005:**
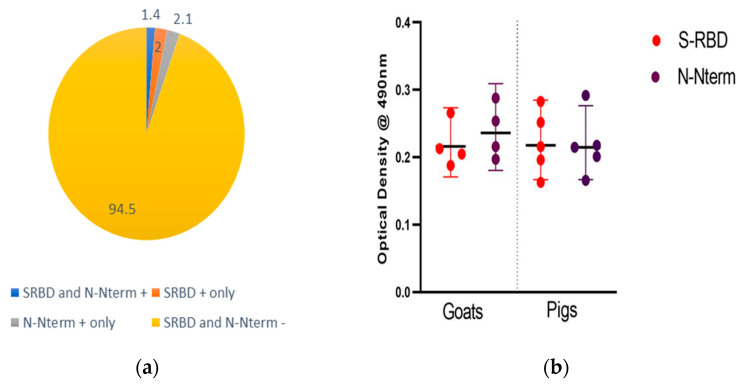
(**a**) Percentage variation of antibody response to SARS-CoV-2 in different animal species; (**b**) range of OD values for SRBD and N N-Term positive animals.

**Figure 6 viruses-15-01187-f006:**
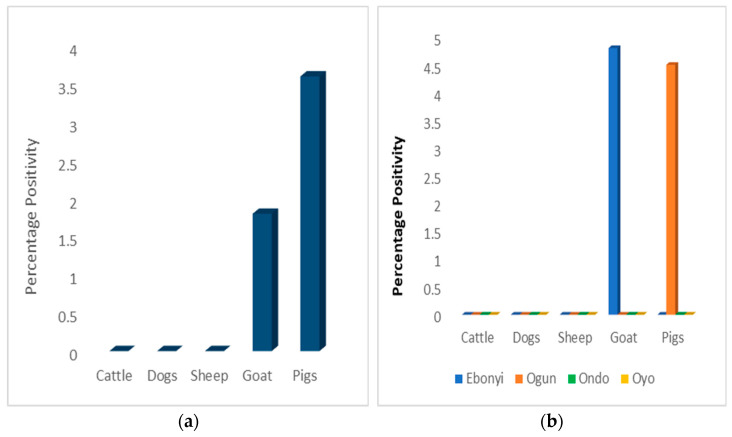
(**a**) Overall SARS-CoV-2 ELISA positivity in animal species; (**b**) comparison of ELISA positivity in animal species by location.

**Figure 7 viruses-15-01187-f007:**
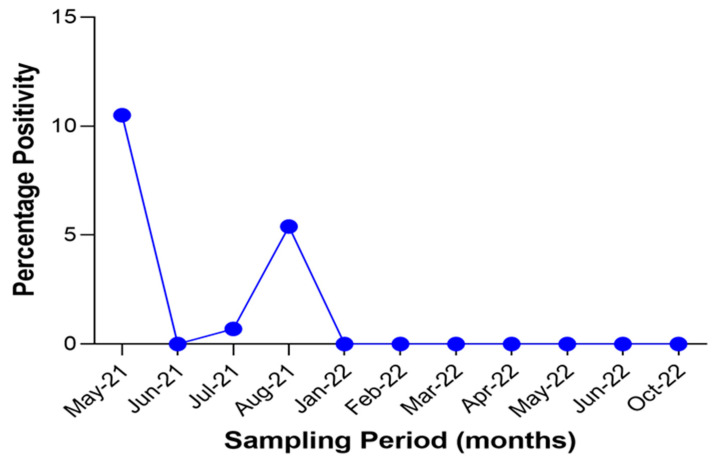
Temporal distribution of SARS-CoV-2 positivity using ELISA in animal species in the sampling locations over the duration of study.

**Table 1 viruses-15-01187-t001:** Distribution of animals sampled by locations.

State	Animal Type	Total Number Analyzed (%)	Number of RT-qPCR Analyzed Samples	Number of ELISA Analyzed Samples
Overall	Cat	2 (0.3)	2	0
	Cattle	24 (3)	24	3
	Chicken	38 (4.8)	38	0
	Dog	172 (21.7)	23	161
	Duck	3 (0.4)	3	0
	Goat	234 (29.6)	77	223
	Lizard	31 (3.9)	31	0
	Pig	147 (18.6)	92	137
	Pigeon	1 (0.1)	1	0
	Sheep	132 (16.7)	56	130
	Turkey	7 (0.9)	7	0
	Total	791 (100)	364	654
Ebonyi	Cat	1 (0.4)	1	0
	Cattle	17 (7)	17	0
	Chicken	9 (3.7)	9	0
	Dog	56 (23)	11	50
	Goat	94 (38.7)	34	83
	Lizard	12 (4.9)	12	0
	Pig	12 (4.9)	3	10
	Sheep	42 (17.3)	19	41
	Total	243	106	184
Ogun	Dog	5 (3.5)	5	5
	Goat	17 (11.8)	5	17
	Pig	116 (80.6)	72	110
	Sheep	6 (4.2)	0	6
	Total	144	82	138
Ondo	Cat	1 (0.3)	1	0
	Cattle	4 (1)	4	0
	Chicken	29 (7.6)	29	0
	Dog	111 (29.1)	17	106
	Duck	3 (0.8)	3	0
	Goat	120 (31.5)	35	120
	Lizard	19 (5)	19	0
	Pig	4 (1)	2	2
	Pigeon	1 (0.3)	1	0
	Sheep	82 (21.5)	35	81
	Turkey	7 (1.8)	7	0
	Total	381	153	309
Oyo	Cattle	3 (13)	3	3
	Goat	3 (13)	3	3
	Pig	15 (65.2)	15	15
	Sheep	2 (8.7)	2	2
	Total	23	23	23

**Table 2 viruses-15-01187-t002:** Comparison of SARS-CoV-2 PCR positivity rates by animal species per state.

State	Animal Species	No. Analyzed	Positive (%)	*p* Value
Overall	Cat	2	1 (50)	1.0
	Cattle	24	17 (70.8)	0.01
	Chicken	38	10 (26.3)	0.01
	Dog	33	17 (51.5)	0.496
	Duck	3	2 (66.7)	0.596
	Goat	77	46 (59.7)	0.01
	Lizard	31	19 (61.3)	0.07
	Pig	92	4 (4.3)	<0.0001
	Pigeon	1	0 (0)	-
	Sheep	56	50 (89.3)	<0.0001
	Turkey	7	1 (14.3)	0.13
	Total	364	167 (45.9)	
Ebonyi	Cat	1	1 (100)	-
	Cattle	17	15 (88.2)	0.02
	Chicken	9	4 (44.4)	0.29
	Dog	11	5 (45.5)	0.33
	Goat	34	20 (58.8)	0.62
	Lizard	12	2 (16.7)	0.001
	Sheep	19	18 (94.7)	0.001
	Pig	3	1 (33.3)	0.56
Ogun	Dog	5	1 (20)	0.23
	Goat	5	1 (20)	0.23
	Pig	72	2 (2.8)	0.07
Ondo	Cat	1	0 (0)	-
	Cattle	4	2 (50)	0.62
	Chicken	29	6 (20.7)	<0.0001
	Dog	17	11 (64.7)	0.91
	Duck	3	2 (66.7)	1.0
	Goat	35	25 (71.4)	0.26
	Lizard	19	17 (89.5)	0.01
	Pig	2	1 (50)	1.0
	Pigeon	1	0 (0)	-
	Sheep	35	32 (91.4)	<0.0001
	Turkey	7	1 (14.3)	0.01
Oyo	Cattle	3	0 (0)	-
	Goat	3	0 (0)	-
	Pig	15	0 (0)	-
	Sheep	2	0 (0)	-

## Data Availability

Not applicable.

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
