# Peer review of "Detection of SARS-CoV-2 in Terrestrial Animals in Southern Nigeria: Potential Cases of Reverse Zoonosis"

_viruses, 2023, doi:10.3390/v15051187_

Round 1

Reviewer 1 Report

In this manuscript the authors (Happi et al.) describe their studies of occurrence of SARS-CoV-2 in domestic and peri-domestic animals in parts of southern Nigeria, with the conclusion that the infections most likely are due to zooanthroponosis (i.e. reverse zoonosis). The manuscript is generally well-written, and within the limitations of the study, the report should be of interest to epidemiologists, veterinarians, veterinary virologists and public health workers.

Having said that, a major limitation of the study is that no attempt at virus isolation was made. While viral genomic RNA may be indicative of infection, it does not reveal whether this is a productive infection or just environmental “sampling” – at least with regards to the nasal swabs. The rectal swabs may be more reliable in this regard, although it also seems surprising that viral RNA survived storage at only -20C, whether still in the collected sample or as purified RNA. Overall the Ct values are relatively high and hence, there may only be very low levels of viral genome material in the samples or it may have degraded due to storage conditions.

It would also be appropriate to provide more information on the households from which the animals were sampled, i.e., number of people and age range, number of animals sampled per household and their age-range, and in the case of cats and dogs, whether they were in- or out-door pets. Were the lizards sampled pets or were they peri-domestic wildlife? It would also be appropriate to provide species name(s) for the lizards.

On what basis was a serum-dilution of 1:100 for the ELISA based? It seem a rather high dilution considering it is well-known that antibody responses to SARS-CoV-2 are rather transient. For example, in ref. 33 the investigators found that animals with a low antibody titer at 2 weeks post infection were negative at 28 days post infection. In other words, if the virus only replicates to low levels and the animal is not clinically affected, then they may quickly turn sero-negative – notably if only tested at a dilution of 1:100.

The manuscript could be further improved by addressing the following:

·       Lines 18-19: this sentence needs rephrasing for improvement of syntax and general clarity.

·       Line 20-21: suggest “However, there is little information from Africa on the spread of the virus among animals in contact with humans and between animals.”

·   Lines 65-66: detection of SARS-CoV-2 in mink is certainly not limited to the USA. Some of the first instances of this happened in Europe, notably Denmark, The Netherlands, France and Poland (see e.g., PMID: 33541485, 33445704, 34126387, 34248922, 36778275).

·       Line 90-91: delete “and other clinical symptoms”. Animals do not have “symptoms” but signs, and in any circumstance the phrase “such as” was used, hence the sentence can end with “and skin lesions”.

·       Line 100: correct to “except for the lizards”.

·       Lines 109 & 291: change ‘nose mask’ to “face mask”.

·       Lines 115-117: please provide manufacturer information for the various PCR kits tested.

·       Line 129: please provide city and country information for Avicenna.

·       Line 141: it is a bit surprising that the investigators did not have access to a conjugated anti-chicken IgY, as these are available from several commercial companies (e.g., Thermo Fischer, Jackson ImmunoResearch, Southern Biotech, Abcam etc.).

·       Line 163: correct to “The animals comprised 11 different …….”

·       Line 275: correct to “for both SARS-CoV-2 by RT-qPCR”

·       Line 293: correct to “laminar flow hood”

·       Lines 330-31: suggest rephrasing the second half of the sentence for clarity.

·       Line 246: correct to “infection rate”

·       Line 248: correct to “proximity to humans”

·       There are a number of inconsistencies in how the references are presented in the Reference list. This should be corrected and standard format as per Instructions to Authors should be followed.

END

Some minor suggestions to improvement of the English is listed in the above.

Reviewer 2 Report

There is evidence that the SARS-CoV-2 pandemic originated in the zoonotic spread of the virus from one or more species of animals to human.  This manuscript explores the interesting scenario of a “reverse zoonotic” spread of the virus from multiple animal species to humans.  In a very broadly based study, almost 800 animals of a wide range of species co-existing in close contact with humans in four different states in Southern Nigeria are examined for evidence for infection by the virus using RT-qPCR and IgG ELISA for the presence of antibodies to the SARS-CoV-2 nucleoprotein and the RBD of the spike protein. 

The main take-aways from this study are that : 1) the virus can infect a wide range of species, encompassing virtually all animal taxa, including cats, dogs, cattle, chickens, ducks, goats, pigs, sheep, lizards, etc.; 2) in this part of the world, as in many others, all of these animals live in close contact with humans either in households or small farms; 3) there is no correlation with detection of the virus and clinical symptoms, i.e., similar to humans, it seems likely that animals with no apparent illness can spread the virus; 4) infectivity rates in animals were much higher in 2021 than in 2022, coinciding with the same trends in humans; and, 5) there was very little virus in both humans and animals in a single state, Oyo State, in 2022.  All of this is interpreted to suggest that there may be multiple cases of reverse zoonosis taking place in this environment, in which humans live in close contact with infected animals.

The scope and breadth of this study are very impressive, especially with respect to the number of animals and the wide range of species evaluated, as well as the opportunity to conduct the study over a period of time that includes a relatively large drop-off in the infection rates in humans, as well as animals.  The significance of the study is considered to be high, as it highlights the importance of monitoring for SARS-CoV-2 across the animal spectrum with an increase portending a potential spillover from any species to humans.  Although previous studies have identified cases of reverse zoonosis, in which SARS-CoV-2 is transmitted from humans to household pets such as dogs and cats, this study is considered only strongly suggestive of a reverse zoonosis, in which the virus is transmitted from humans to multiple species living in close contact.  A correlation between infection of animals and humans does not constitute definitive proof of reverse zoonosis. All of the findings are consistent with these events, but do not conclusively prove them.  It seems that a definitive link between infected animals and humans would be best established by the isolation of a genotypically conserved virus in one or more previously uninfected humans closely following exposure to an animal previously infected with the same virus.

All of this being said, in my opinion, the authors should simply back off a bit from their conclusion that these human infections are evidence of reverse zoonosis.  For example, in the title itself, this would entail the simple insertion of the word “potential” before the word “cases”.  Indeed, in the conclusion section, they state that: “The proximity between humans and animals in this setting as well as temporal correlation of positivity rates strongly suggest a human-animal transmission.”  Although I suspect they meant animal-human transmission here, this thinking should be expressed throughout the manuscript.  In addition, another potentially important aspect of these findings that I feel is not stressed in the manuscript is the potential for the virus to mutate to a more dangerous form in one of these animals, which could be disastrous.

English language is acceptable.

Reviewer 3 Report

SUMMARY:

.           This paper describes a large field sampling study of animals to detect the prevalence of SARS-CoV2 in terrestrial animals co-habiting with humans in southern Nigeria. Oral, rectal and plasma samples were collected from a variety of species and analysed in this study. Swabs of oral and rectal/cloacal compartments in animals were subjected to RNA extraction and screened for the presence of SARS-CoV2 RNA via RT-qPCR. Plasma samples from 5 species of domestic animals were subjected to IgG ELISA to detect the presence of neutralizing antibodies towards SARS-CoV2 spike protein and N protein.

Specific comments below:

.           Abstract Line 18: “Since SARS-CoV-2 caused the zoonosis COVID-19 pandemic...” Please change to: “Since SARS-CoV-2 caused the zoonotic COVID-19 pandemic...”

.           Abstract Line 18: “…, pets and farm animals in contact with SARS-CoV-2 positive humans suggested the occurrence of reverse zoonosis.” Please reword this sentence, there is no statement in this sentence saying that pets and farm animals are positive for SARS-CoV-2. Which I think is what the authors are trying to say. Please state this explicitly

.           Abstract Line 21: “Therefore, this study aimed to investigate SARS-CoV-2 in various…” please change to: “Therefore, this study aimed to investigate prevalence of SARS-CoV-2 in various…”

.           Abstract Line 26: “SARS-CoV-2 IgGs were detected in only goats…” Please change to ““SARS-CoV-2 IgGs were detected only in goats…”  

.           Abstract Line 29: “SARS-CoV-2 natural infection in poultry,…” Please change to: “natural SARS-CoV-2 infection in poultry,…”

.           Abstract Line 29: Sentence beginning with “The close human-animal interaction…” consider re-wording this sentence. It’s very hard to understand. My suggestion is to break it up into 2 sentences: “… suggest ongoing reverse zoonosis, the behavioral factor of trans-” change to: “… suggests ongoing reverse zoonosis. This reverse zoonosis, the behavioral factor of trans-…”

.           Line 48-49: “However, it is speculated that many COVID-19 cases went unreported [11,12]”. Please clarify if these unreported cases are unreported globally or unreported for Nigeria specifically

.           Line 52: “Southern Nigeria has been categorized as a high risk and hotspot zone…” Please delete the word “hotspot” in this sentence

.           Line 54: “Most of the first set of hospitalized COVID-19 cases were...” please consider changing to: “A majority of the initial cases of COVID-19 requiring hospitalization were...”

.           Line 67: “… farm animals is not a new phenomenon.” Please change the word “a new” to “an uncommon”.  Human cohabitation is not a new phenomenon in many parts of the world, in fact cohabiting with animals is probably an old phenomenon. Not cohabiting with animals and living in separate dwellings is probably a newer phenomenon, but in many developed nations cohabiting with animals is uncommon.

.           Line 68:  “… ample opportunity for the exposure of…” please delete the word “the” from this sentence

.           Line 70-73: “However, there are gaps in information on the extent of SARS-CoV-2 spread to other animals in Africa, and this is vital in guiding the adoption of an effective public health approach for more sustainable and holistic measures of control and of monitoring the virus spread.” This sentence is too long and is difficult to read. Please consider breaking it up into separate sentences that are easier to understand

.           Line 93: Figure 1 general suggestion: total number of samples from each of the sampling sites could be included for each of the states

.           Line 96: “…animals found living in close contact with humans…” General comment it would be good to expand this definition of close contact in terms of time spent in the presence of humans? Did animals share the same dwellings?

.           Line 116: “GeneFinderTM COVID-19 Plus RealAmp kit, the AviMol Dri Kit, and the AllplexTM” please provide supplier information for these kits. Also please change all trademark letters eg: “GeneFinderTM” to the correct trademark symbol: “GeneFinder™”

.           Line 118: General comment on the validation process. Please explain what positive or negative controls were used to validate your kit and why you chose AviMol Dri Kit. It is not very clear.

.           Line 125: Grammar: “The cycling condition includes…” please change to: “The cycling conditions include…”

.           Line 133: “…CT values between 38 and 40 were rerun and called positive if…” please replace the word “called” to “considered”

.           Line 134: Sentence beginning with: “An animal was called positive please replace the word “called” to “considered”

.           Lines 135 -138: “Given that the RT-qPCR analysis was done during the period we recorded very few COVID -19 cases in Nigeria and other countries in the world (October 2022), results were considered only when the negative extracts and negative controls showed no amplification.” Please rewrite this sentence more simply. It is very hard to understand what the authors are saying here. I believe the authors are trying to say that RT-qPCR results were only true if the negative controls showed no amplification. I am unsure why Nigeria’s human COVID -19 cases at the time the analysis was conducted as any bearing on the result of this separate study. The outcome of this RT-qPCR run is not dependent on human COVID -19 case-loads.

.           Line 158: Section 2.6. Statistical Analysis, there is no mention of a specific type of statistical analysis performed on these results. Was it a two paired students T test? Annova?

.           Line 190: Figure 2. Comparison of PCR Ct Values in different animal species. General comment: Please provide a short explanation on what is observed in the graph, eg. No difference between oral and rectal swab in the presented species with the exception of cattle. Also mention what statistical test was performed.

.           Line 196: Figure 3. Comparison of SARS-CoV-2 PCR positivity rates in animals by location in Nigeria. General comment, again please provide a short explanation of what is being observed in the graph, also there are no error bars on these graphs.

.           Line 199-201: General comment: positivity for SARS-CoV-2 as determined by which method? Please explicitly state this.

.           Line 212: “SARS-CoV-2 positivity was non-signif-…” Please change the word “non-” to “not”

.           Typo Line 235: “The cut-off O.D val-“ please change to : “The cut-off OD val-“

.           Typo Line 244: “Range of OD values for SRBD and N N-Term positive animals .” remove space before full stop.

.           Lines 246-248 and lines 250-251: given that the ELISA results between Ebonyi and Ogun states are not significantly different I don’t think the authors can claim that one state was higher than the other. Again authors need to state what type of statistical analysis was performed to come to this conclusion.

.           Line 258: Figure 6. General comments, please include error bars on the graphs

.           Line 274: If the data is available please expand on these results. This might be interesting in terms of which epitope animals mount their immune responses to in comparison to humans

.           Line 278: “The origination of SARS-CoV-2…” please change the word “origination” to “origins”

.           Line 282: “…in close contact with humans based on the community…” please delete the word “the”

.           Line 299-300: “Overall, there was no significant difference in SARS-CoV-2 positivity rates and cycle 299 threshold values between oral and rectal swab samples among animals.” This statement contradicts the data in figure 2 where there is a significant difference between the oral and rectal samples of cattle

.           Line 317-319: general comment on the discussion about pigeons. Perhaps this sample should be excluded from the study and subsequent discussions as there is only one sample to speak of?

.           Line 346: Grammar “… were relatively lower compared” change to: “was relatively low in comparison…”

.           Line 347: “… that there was a high infection of SARS-CoV-2 in animals…” please change to: “… that there was a high infection rate for of SARS-CoV-2 in animals…”

.           Line 348: Grammar “… due to their proximity of humans…” change to: “… due to their proximity to of humans…”

.           Line 360: “SARS-CoV-2infected cases detected in animals” please change to “SARS-CoV-2 infectionsed cases detected in animals”

.           Line 369: line beginning with “Although these were experi-“ please change to “Although these animals were experi-“

.           Line 391: “once a pathogen crosses an animal to human,” please change to: “once a pathogen crosses from an animal to human,”

General comment for the discussion section. Was there any sequencing done in the samples collected to look at what variant of SARS-CoV-2 was infecting the animals and how closely it matched with what was happening in human populations at the same time and place? Sequences from animals that are a genetic match to the human variant of SARS-CoV-2 would be conclusive evidence of human to animal transmission. 

English is generally fine, there were some sentences that had poor structure and could do with some simplification to get the message accross.

Round 2

Reviewer 1 Report

The authors are commended for close attention to this reviewer's comments and suggestions. The manuscript reads well and should be of interest to many investigators in the SARS-CoV-2 and "One Health" fields.

English Language is overall asseptable.

Author Response

Dear Reviewer 1,

Thank you very much for your valuable advice. We re-read the manuscript and have tried to edit the text. Please find the revised version 2 of the manuscript. We hope that the English language is much acceptable.

Yours sincerely,

Anise Happi

Reviewer 3 Report

Specific comments below for version 2:

.           Line 192: “A total of 791 animal samples were analyzed”. I believe that there would be a blood sample, as well as an oral and rectal swab for nearly every animal sampled. And this sentence describes the number of animals sampled rather than the number of samples analysed. Therefore you should consider changing this line to “Samples from A total of 791 animals samples were analyzed”. Or consider adding the total number of samples instead.

.           Line 217: “… the mean CT value for…” please be consistent and change to Ct, which is what is how cycle threshold is abbreviated throughout the text. Also fix this in line 152.

.           Line 394: “… our argument that the SARS-CoV-2 infectioned cases detected…” please either use infection or cases, both words are not needed here and can be confusing.

General comment for the Statistical Analysis Section 2.6.

Thank you for adding these crucial details.

Much improved quality now that the edits from the reviewers have been incorporated
